# Antidiabetic and Antihyperlipidemic Effects of Sulphurenic Acid, a Triterpenoid Compound from *Antrodia camphorata*, in Streptozotocin-Induced Diabetic Mice

**DOI:** 10.3390/ijms20194897

**Published:** 2019-10-02

**Authors:** Cheng-Hsiu Lin, Li-Wei Hsiao, Yueh-Hsiung Kuo, Chun-Ching Shih

**Affiliations:** 1Department of Internal Medicine, Fengyuan Hospital, Ministry of Health and Welfare, Fengyuan District, Taichung 42055, Taiwan; keny71@pchome.com.tw; 2Division of Endocrinology and Metabolism, Chang Bing Show Chwan Memorial Hospital, Changhua 505, Taiwan; kmccenor@gmail.com; 3Department of Chinese Pharmaceutical Sciences and Chinese Medicine Resources, China Medical University, Taichung 40402, Taiwan; kuoyh@mail.cmu.edu.tw; 4Graduate Institute of Biotechnology and Biomedical Engineering, College of Health Science, Central Taiwan University of Science and Technology, No.666 Buzih Road, Beitun District, Taichung 40601, Taiwan

**Keywords:** diabetes, streptozotocin, sulphurenic acid, glucose transporter 4, Akt phosphorylation

## Abstract

The present study was designed to evaluate the protective effect of sulphurenic acid (SA), a pure compound from *Antrodia camphorata*, on diabetes and hyperlipidemia in an animal model study and to clarify the underlying molecular mechanism. Diabetes was induced by daily 55 mg/kg intraperitoneal injections of streptozotocin (STZ) solution over five days. Diabetic mice were randomly divided into six groups and orally gavaged with SA (at three dosages) or glibenclamide (Glib), fenofibrate (Feno) or vehicle for 3 weeks. Our findings showed that STZ-induced diabetic mice had significantly increased fasting blood glucose, glycated hemoglobin (HbA1_C_), plasma triglyceride (TG), and total cholesterol (TC) levels (*p* < 0.001, *p* < 0.001, *p* < 0.001, and *p* < 0.05, respectively) but decreased blood insulin, adiponectin, and leptin levels compared to those of the control group (*p* < 0.001, *p* < 0.001, and *p* < 0.001, respectively). Administration of SA to STZ-induced diabetic mice may lower blood glucose but it increased the insulin levels with restoration of the size of the islets of Langerhans cells, implying that SA protected against STZ-induced diabetic states within the pancreas. At the molecular level, SA treatment exerts an increase in skeletal muscle expression levels of membrane glucose transporter 4 (GLUT4) and phospho-Akt to increase the membrane glucose uptake, but the mRNA levels of PEPCK and G6Pase are decreased to inhibit hepatic glucose production, thus leading to its hypoglycemic effect. Moreover, SA may cause hypolipidemic effects not only by enhancing hepatic expression levels of peroxisome proliferator-activated receptor α (PPARα) with increased fatty acid oxidation but also by reducing lipogenic fatty acid synthase (FAS) as well as reducing mRNA levels of sterol regulatory element binding protein (SREBP)1_C_ and SREBP2 to lower blood TG and TC levels. Our findings demonstrated that SA displayed a protective effect against type 1 diabetes and a hyperlipidemic state in STZ-induced diabetic mice.

## 1. Introduction

Type 1 diabetes (T1D), also known as juvenile diabetes or insulin-dependent diabetes, is a form of diabetes in which very little or no insulin is produced by the pancreas. Before treatment, this leads to high blood glucose levels in the body. The cause of type 1 diabetes is believed to involve a combination of genetic and environmental factors. The mechanism includes destruction of the insulin-producing beta cells in the pancreas. Diabetes is diagnosed by testing the level of sugar or glycated hemoglobin (HbA1_C_) in the blood [1]. Type 1 diabetes makes up an estimated 5%~10% of all diabetes cases [2]. Type 1 diabetes is associated with many complications, and those are considered the causes of morbidity and mortality in patients with diabetes. There is no known way to prevent type 1 diabetes. Management with insulin is required for survival.

*Antrodia camphorata* (Polyporaceae, Aphyllophorales) is a previous edible fungus and has been used as a folk remedy in Taiwan. Because it only grows in the inner heartwood wall of the endemic evergreen *Cinnamomum kanehirai*, it is rare and expensive. It consists of numerous bioactive compounds that have been demonstrated to improve health and ameliorate various diseases. The mycelia of *A. camphorata* display anticancer activity, liver protection, immunomodulation, antioxidant and scavenging free radicals, and anti-inflammatory activity; its broth filtrate showed anticancer, and the fruiting body displayed anticancer, liver protection, and immunomodulation activities [3]. Evidence has shown that the solid culture of the fruiting body and the filtrate in submerged culture have a hepatoprotective effect and antioxidant activity [4,5]. Our recent studies demonstrated that ergostatrien-3β-ol (EK100) [6], dehydroeburicoic acid (TR2), euricoic acid (TR1) [7,8,9,10], and antcin K [11] from *A. cinnamomea* exhibited antihyperglycemic and antihyperlipidemic activity. Nevertheless, the antidiabetic and antihyperlipidemic potential activities of sulphurenic acid (24-methylenelanosta-8-ene-3β,15α-diol-21-oic acid, 10; TR3; SA) (Figure 1) from *A. camphorata* remain unknown in streptozotocin (STZ)-induced diabetic mice.

Streptozotocin (STZ) is one of the most universally used diabetogenic agents to induce diabetes in experimental animals [12]. It is prominent for its selective pancreatic β-cell cytotoxicity and has been extensively used to induce insulin-dependent diabetes mellitus or type 1 diabetes [13,14]. Streptozotocin is a nitric oxide donor, and nitric oxide can cause the destruction of pancreatic islet cells. Streptozotocin by itself was demonstrated to generate reactive oxygen species (ROS), which contributed to DNA fragmentation and evoked other deleterious changes within the pancreatic tissue [15,16]. Multiple low dose (MLD)-STZ injections (35–55 mg/kg bodyweight for 4–5 consecutive days) are often used to model destruction of pancreatic β cells and hyperglycemia and can be used as a model for insulin-dependent diabetes (IDDM), which is accompanied by a 70% reduction in the islet per pancreas area [17].

Glucose transporter 4 (GLUT4) is the major insulin-regulated glucose transporter expressed mainly in the skeletal muscles and adipose tissues [18,19]. Skeletal muscle is proposed to be the primary site of whole-body insulin-mediated glucose uptake [20,21,22]. Insulin stimulates glucose uptake in these cells primarily by inducing the net translocation of GLUT4 from the intracellular storage sites to the plasma membrane.

There are two major cellular mechanisms to account for the promoted translocation of GLUT4 to the plasma membrane: insulin signaling through the phosphatidylinositol 3′ kinase (PI3-kinase)/Akt pathway and the AMP-activated protein kinase (AMPK) pathway [23,24,25].

Glibenclamide (Glib) is a second generation analog of sulfonylureas. Glibenclamide is an oral hypoglycemic drug that stimulates pancreatic beta cells to secrete insulin [26]. The mechanism of glibenclamide is to stimulate insulin secretion from the islet β-cell under the preliminary conditions that it must still have a part of its storage function and that the pancreas completely or almost completely has no insulin-secretion action. Glibenclamide enhances insulin action in the cells in culture and stimulates the synthesis of glucose transporters [27]. Sulfonylureas have also been shown to suppress hepatic gluconeogenesis [28].

The present study was designed to evaluate the potential activity of SA on the regulation of blood glucose and lipid metabolism and to further clarify the underlying molecular mechanism of SA. Phosphorylation of Thr^172^ of α subunits is essential for AMPK activity [29]. Therefore, the aim of the present study is to assess whether the blood glucose and lipids were modulated in SA-treated STZ-induced diabetic mice and to compare these changes with clinical drugs, including Glib and Feno. Glibenclamide is a sulfonylureas that causes hypoglycemia by stimulating insulin release from pancreatic β cells [26]. Fenofibrate, an agonist of peroxisome proliferator-activated receptor α (PPARα) now used for the treatment of hypertriglyceridemia, can cause hypolipidemia [30]. Phosphoenolpyruvate carboxykinase (PEPCK) and glucose-6-phosphatase (G6Pase) are the key rate-limiting enzymes of gluconeogenesis [31]. Based on one of the possible mechanisms, we also investigated whether SA regulated the expression of genes involved in antidiabetes, lipogenesis, and fatty acid oxidation, including PEPCK, G6Pase, peroxisome proliferator-activated receptor α (PPARα), and lipogenic fatty acid synthase (FAS) in STZ-induced diabetic mice.

## 2. Results

### 2.1. Expression Levels of Phosphorylated Akt in a Cell Line

The experiment was performed in a cell line. As shown in Figure 2A,B, our findings showed that SA activates Akt in a time-dependent manner. The expression levels of phosphorylated Akt reached a maximum level at 60 min of SA treatment, which is comparable to that of insulin.

### 2.2. Treatments in STZ-induced Diabetic Mice

#### 2.2.1. Body Weight, Food Intake, and Relative Tissue Weights

The body weight average of all mice when they entered the animal room was 15.81 ± 0.42 g, and one week of acclimatization followed. After STZ induction, the body weight of the normal control (CON) group was 20.97 ± 0.74 g, and that of STZ mice was 19.19 ± 0.17 g (*p* < 0.05). Then, the STZ mice were subdivided into five groups, followed by treatment with vehicle (which was represented as the STZ group), SA1, SA2, Glib, or Feno for 3 weeks. At the end of the experiment, we found that the final body weight of the STZ mice was lower than that of the CON group (*p* < 0.05), and there was no significant difference in the final body weight between the SA-, Glib-, or Feno-treated mice and the vehicle-treated STZ mice (Figure 3A). The STZ-induced mice consumed more food intake than the CON mice (*p* < 0.05), and there was no difference in food intake in the SA-, Glib-, and Feno-treated mice compared to that of the vehicle-treated STZ mice (Figure 3B).

The STZ induction caused a significant decrease in the relative weights of epididymal white adipose tissue (WAT), retroperitoneal WAT, visceral fat, and skeletal muscle (*p* < 0.001, *p* < 0.001, *p* < 0.001, and *p* < 0.001, respectively) but an increase in the relative weights of liver tissues compared with those of the CON group (*p* < 0.001) (Figure 3C,D). There were no significant differences in the relative weights of epididymal WAT, mesenteric WAT, retroperitoneal WAT, visceral fat, skeletal muscle, liver tissues, and brown adipose tissue (BAT) between the SA1-, SA2-, SA3-, or Glib-treated STZ mice and the vehicle-treated STZ mice. Treatment with Feno significantly decreased the relative weights of the epididymal WAT (*p* < 0.05) but enhanced the relative weights of skeletal muscle and liver tissues compared with the STZ-treated mice (*p* < 0.01 and *p* < 0.001, respectively) (Figure 3D).

#### 2.2.2. Blood Metabolic Parameters

The STZ-induced mice displayed a significant increase in the levels of blood glucose and HbA1_C_ compared with the CON group (*p* < 0.001 and *p* < 0.001, respectively) (Figure 3E,F). Administration of SA1, SA2, or SA3 lowered the blood glucose levels (*p* < 0.05, *p* < 0.05, and *p* < 0.001, respectively) compared with levels in the STZ group (Figure 3E). Treatment with SA3 reduced the blood HbA1_C_ levels (*p* < 0.05) compared with levels in the STZ group (Figure 3F). The STZ-induced diabetic mice had markedly elevated the plasma triglyceride (TG) and total cholesterol (TC) levels compared with those of the CON group (*p* < 0.001 and *p* < 0.05, respectively) (Figure 3G,H). Treatment with SA1, SA2, SA3, Glib, or Feno significantly lowered the plasma TG levels compared with levels in the STZ group (*p* < 0.05, *p* < 0.01, *p* < 0.001, *p* < 0.05, and *p* < 0.001, respectively) (Figure 3G). Treatment with SA1, SA2, SA3, Glib, or Feno displayed a decrease in plasma TC levels compared with that of the STZ group (*p* < 0.05, *p* < 0.01, *p* < 0.01, *p* < 0.05, and *p* < 0.05, respectively) (Figure 3H). The STZ induction exhibited a dramatic decrease in blood insulin levels compared with that of the CON group (*p* < 0.001), and treatment with SA1, SA2, or SA3 significantly enhanced blood insulin concentrations compared with the concentration in the STZ group (*p* < 0.05, *p* < 0.01, and *p* < 0.01, respectively) (Figure 3I).

STZ induction significantly reduced the blood levels of adiponectin and leptin compared with those of the CON group (*p* < 0.001 and *p* < 0.001, respectively) (Figure 3J,K). Treatment with SA2, SA3, Glib, or Feno elevated adiponectin (*p* < 0.01, *p* < 0.01, *p* < 0.05, and *p* < 0.001, respectively) levels in the blood (Figure 3J). Administration of SA3 and Feno markedly enhanced leptin levels compared with levels in the STZ group (*p* < 0.001 and *p* < 0.001, respectively) (Figure 3K). The STZ induction significantly increased the plasma free fatty acid (FFA) levels compared with that of the CON group (*p* < 0.001) (Figure 3L). Treatment with SA2, SA3, Glib, or Feno significantly decreased the plasma FFA levels (*p* < 0.001, *p* < 0.001, *p* < 0.001, and *p* < 0.001, respectively) (Figure 3L).

#### 2.2.3. Histology

The STZ-induced diabetic mice displayed slight ballooning of hepatocytes compared with that of the CON group, and treatment with SA1, SA2, SA3, Glib, or Feno displayed no ballooning phenomenon (Figure 4A). The islets of STZ-induced diabetic mice exhibited retraction from their classic round shape compared to the control islets, and following treatment with SA1, SA2, and SA3, they displayed an improvement in the size of islets and less degeneration within the pancreas (Figure 4B). As shown in Figure 4C, STZ-induction leads to a decrease in the average areas of islets of Langerhans as compared with the CON group (*p* < 0.001). SA1, SA2, and SA3 treatment lead to an increase in the average areas of islets of Langerhans as compared with the vehicle-treated STZ mice (*p* < 0.001, *p* < 0.001, and *p* < 0.001, respectively). There exists no significant difference in the Glib- and Feno- treated STZ mice as compared with the vehicle-treated STZ mice in the average areas of islets of Langerhans. Immunostaining results for insulin (brown) and glucagon (green) are shown in the Figure 4D, treatment with SA3 significantly increased the insulin levels and beta cell numbers.

#### 2.2.4. Hepatic Targeted Gene mRNA Levels

As shown in Figure 5, the mRNA levels of glucose 6-phosphatase (G6Pase), phosphoenolpyruvate carboxykinase (PEPCK), sterol regulatory element binding protein 1c (SREBP1c), and SREBP2 were lower in the STZ group than those in the CON group (*p* < 0.001, *p* < 0.001, *p* < 0.001, and *p* < 0.001, respectively). Administration of SA2 or SA3 reduced the mRNA levels of G6Pase and PEPCK. Treatment with SA1, SA2, SA3, Glib, or Feno decreased the mRNA levels of SREBP1c and SREBP2 compared to the levels in the STZ group (Figure 5A,B).

#### 2.2.5. Targeted Gene Expression Levels in Different Tissues

As shown in Figure 6, the membrane expression of GLUT4 was lower in the STZ group than that in the CON group in skeletal muscles (*p* < 0.001). Administration of SA1, SA2, SA3, Glib, or Feno significantly enhanced the membrane expression of GLUT4 compared to expression in the STZ group (*p* < 0.001, *p* < 0.001, *p* < 0.001, *p* < 0.001, *p* < 0.001, and *p* < 0.001, respectively). The expression levels of p-AMPK/t-AMPK and p-Akt/t-Akt were lower in the STZ group than those in the CON group in the skeletal muscle (*p* < 0.001 and *p* < 0.001, respectively), and treatment with SA1, SA2, SA3, Glib, or Feno markedly enhanced the expression levels of p-AMPK/t-AMPK while SA1, SA2, or SA3 markedly enhanced the expression levels of p-Akt/t-Akt compared to those of the STZ group (Figure 6A,B).

As shown in Figure 7, the expression levels of p-Akt/t-Akt were lower in the STZ group than that in the CON group in the liver tissues (*p* < 0.001). The SA1-, SA2-, or SA3-treated groups had markedly enhanced expression of p-Akt/t-Akt compared to that of the STZ group (*p* < 0.05, *p* < 0.001, and *p* < 0.001, respectively) (Figure 7A,B). The hepatic expression levels of phospho-forkhead transcription factor FoxO1 (phospho-FoxO1)/total-FoxO1 (p-FoxO1/t-FoxO1) were lower in the STZ group than those in the CON group (*p* < 0.001). Treatment with SA1, SA2, or SA3 increased the hepatic expression levels of p-FoxO1/t-FoxO1 compared to those in the STZ group (Figure 7A,B). The hepatic expression levels of p-AMPK/t-AMPK and PPARα were significantly lower in the STZ group than those in the CON group, and treatment with SA1, SA2, SA3, Glib, or Feno increased the hepatic expression levels of p-AMPK/t-AMPK and PPARα compared to those in the STZ group (Figure 7C,D). The hepatic expression levels of FAS and PPARγ were higher in the STZ group than levels in the CON group. Treatment with SA1, SA2, SA3, Glib, or Feno significantly reduced the hepatic expression levels of FAS, and SA2, SA3, Glib, or Feno decreased the expression levels of PPARγ compared with levels in the STZ group (Figure 7C,D).

## 3. Discussion

The present study is designed to investigate the antidiabetic and antihyperlipidemic activity of SA, a triterpenoid compound of *A. camphorata* and its possible molecular mechanism in STZ-induced diabetic mice. In the in vitro experiment, SA displayed a significant increase in Akt phosphorylation in a dose-dependent manner, and the expression levels of p-Akt approached the maximum at 30 to 60 min following treatment with SA in C2C12 myotubes, implying that SA exhibits increased Akt pathway phosphorylation and has an action similar to that of insulin (Figure 2). However, its antidiabetic effect remains poorly understood. Too determine whether SA could display glucose-lowering effects, we assessed its antidiabetic activity using an animal model, namely, STZ-induced diabetic mice, and compared SA with the clinical drug Glib. Although Glib is not used for IDDM patients, Glib may stimulate insulin release [26] in MLD-STZ-induced diabetic mice, with a 70% reduction in islets per pancreas [17], although some β-cell area still exists. The STZ-induced diabetic model has been used as a type 1 diabetic model [13,14]. Therefore, the STZ mouse model was chosen to address antidiabetic activities and to clarify the possible mechanisms of SA. Collectively, the data presented here demonstrated that SA displays antidiabetic activity in type 1 diabetic mice (Figure 8).

Multiple low dose (MLD)-STZ injections are often used to model the destruction of pancreatic β cells and hyperglycemia and can be used as a model for insulin-dependent diabetes (IDDM) [17]. Our findings were in agreement with earlier observations that MLD-STZ injections may lead to hyperglycemia evident in C57BL6J mice [17], indicating that STZ induction displayed β cell mass loss and then insulin insufficiency, leading to hyperglycemia. STZ induction leads to an increase in the level of glycosylated hemoglobin. The level of glycosylated hemoglobin is a marker related to glycemia during the previous 2–3 month period, implying oxidation and damage in tissues [32,33], showing that oxidative damage exists, continuing and reinforcing the cycle of oxidative stress and damage [33]. STZ-induced diabetes mellitus is characterized by a severe loss in body weight [34]. The decrease in body weight is due to both the loss and degradation of structural proteins [35] and the altered carbohydrate metabolism [36]. Our findings may be consistent with earlier observations [34] that STZ induction leads to a decrease in both body weight and blood insulin levels. In addition to hyperglycemia and hypoinsulinemia, our findings showed that reduced body weight and increased levels of blood HbA_1C_ further confirm that the STZ-induced diabetic animal model was successfully established.

Afterward, administration of SA3 for 3 weeks to STZ-induced diabetic mice significantly decreased the blood glycated hemoglobin (HbA1_C_) levels (Figure 3F), indicating that SA3 may prevent oxidative damage caused by the glycation reaction in diabetic conditions. One of the problems for the biochemical examination especially HbA1_C_ level is its small blood volume. Our results showed that HbA1_C_ levels were significantly elevated in STZ-induced diabetic mice (6.0%) as compared with the CON group (4.0%). According to a piece of study, HbA1c usually does not change until 5 weeks of treatment in mice [37]. At the end of this experiment, our mice age is about 11-week age. An explanation for this is due to the different induced-animal models, the different STZ dosage (the previous study used 90 mg/kg; ours are 55 mg/kg of STZ for 5 days), and the different mice strain (theirs is ddY mice). Although treatment with SA1, SA2, and SA3 significantly lowered the blood glucose levels. Nevertheless, there is no significant statistical difference in HbA1_C_ levels between SA1- and SA2- treated STZ mice and vehicle-treated STZ mice, and merely SA3 treatment (5.1%) decreased HbA1_C_ levels as compared with the vehicle-treated STZ mice (6.0%). An explanation is that the different mice strain, the dosage of STZ, mice age, and 3-week SA treatment are not enough to display a decrease in HbA1_C_ levels since hemoglobin is damaged and need time for recovery. Our results indicated that SA may lower the blood glucose levels and raise the blood insulin concentration in STZ-induced diabetic mice (Figure 3E,I). Moreover, SA treatment induced an increase in the size of pancreatic islets compared with the size in vehicle-treated diabetic mice (Figure 4B,C). Morphological and immunohistochemical staining examination revealed that SA treatment after STZ induction displayed an increase in the size and number of Langerhans islet cells within the pancreas (Figure 4B,C) and an evident increase in insulin expressing beta-cells (Figure 4D), implying that SA improve insulin secretion. Taken together, our findings indicate that SA may stimulate the Akt signaling pathway to a similar extent as insulin in in vitro studies (Figure 2), and in animal studies, SA exerts its hypoglycemic effect by acting as an insulin secretagogue from the residual β-cells after destruction by STZ and subsequent regeneration and/or antioxidant activities by HbA1_C_ levels, as well as within the pancreas.

Acute STZ injection has been performed to clarify cellular oxidative damage since it produces reactive oxygen species and decreased antioxidant enzymes (including activities such as superoxide dismutase, catalase, and glutathione peroxidase) in the pancreas, which has been identified as the major deterioration within the pancreas [38]. A previous study showed that diabetic islets displayed retraction from their classic round shape compared to that of control islets [39]. Antioxidants have been suggested to afford protection to the pancreas against oxidative stress in diabetes mellitus [40]. By histological examination, STZ-induced diabetic mice showed shrunken islets of Langerhans displaying degenerative and vacuolative changes, and mice treated with SA following STZ displayed an increase in the size and number of islets of Langerhans cells with less distorted architecture within the pancreas (Figure 4B), suggesting an antioxidant effect within the β-cells against STZ-induced oxidative stress within the pancreas. There is one possibility that the antioxidant action of SA may be ameliorated by the reduction in reactive oxygen species in STZ-injured beta cells.

Insulin and phosphorylation of AMPK are two major mechanisms that account for reduced mRNA levels and gene expression of phosphoenolpyruvate carboxykinase (PEPCK) and glucose-6-phosphatase (G6 Pase) [41]. The forkhead transcription factor forkhead box O1 (FoxO1) plays a crucial role in the regulation of the effect of insulin on hepatic gluconeogenesis. When the blood glucose level is high, the pancreas releases insulin into the bloodstream. Insulin then causes the activation of PI3K, which subsequently phosphorylates Akt. Akt then phosphorylates FoxO1 [42]. The phosphorylation of FoxO1 is irreversible; this prolongs insulin’s inhibitory effect on glucose metabolism and hepatic glucose production. Transcription of G6Pase subsequently decreases, which consequently decreases the rate of gluconeogenesis and glycogenolysis [43]. Insulin may inhibit this pathway by decreasing the transcription of G6 Pase. Insulin inhibits gluconeogenesis via Akt-dependent phosphorylation of forkhead transcription factor Foxo1 (FoxO1), which, in turn, inhibits PEPCK and G6 Pase gene transcription [44]. Our findings show that SA increases blood insulin levels and hepatic expression levels of p-Akt and p-FoxO1, which may result in a decrease in the mRNA levels of G6Pase and PEPCK and consequently inhibit hepatic glucose production with blood glucose-lowering effects. The beneficial antidiabetic effect of SA may be due to an elevation of blood insulin that partly acts via insulin-Akt-FoxO1 to reduce the mRNA levels of G6 Pase and PEPCK, which has a suppressive effect on hepatic glucose production, thus contributing to a hypoglycemic effect.

Skeletal muscle is proposed to be the primary site of whole-body insulin-mediated glucose uptake [20,21,22]. Therefore, this study was designed to evaluate the expression levels of membrane GLUT4 in the skeletal muscle of SA-treated STZ-diabetic mice and the performance of many measurements of targeted gene protein expression in different tissues. Our findings that the expression levels of membrane GLUT4 in skeletal muscles are reduced in STZ-induced diabetic mice compared with the levels in the CON group are in agreement with an earlier observation [45]. SA treatment displayed an increase in the skeletal muscular expression levels of membrane GLUT4 compared with the vehicle-treated STZ group. Moreover, following treatment with SA, blood insulin levels were enhanced, and expression levels of both p-Akt/t-Akt and p-AMPK/t-AMPK in skeletal muscles were also markedly elevated, implying that SA increased membrane glucose transport activity by stimulating the insulin-Akt and/or AMPK activation pathways, further contributing to this reduction in blood glucose levels. The net effect of SA displays antidiabetic activity in mouse models of type 1 diabetes mellitus. Our data suggest that SA may alleviate type 1 diabetes symptoms, or may be used in combination with insulin, and it will not be a sole treatment.

Collectively, there are differences between Glib and SA treatment on the glycemic effects in MLD-STZ-induced diabetic mice. With chronic administration, circulating insulin levels decreased, and despite the increase in membrane glucose uptake, Glib had no efficacious effects on the blood glucose and HbA1_C_ levels or the size of the residual islets of Langerhans cells. The explanation for this is not clear, but it may relate to reduced residual islet cells that impair insulin secretion and to the fact that the insulin level is too low to inhibit hepatic glucose production. All of our results indicated that there is no beneficial effect of Glib therapy on glycemic control in STZ-induced diabetic mice, and a 3-week duration of treatment with Glib in type 1 DM has not been suggested to have a good response.

Aside from the hypoglycemic activity of SA, another objective of this study was to clarify the antihyperlipidemic activities and underlying molecular mechanisms of SA. Hypertriglyceridemia and hypercholesterolemia have been shown to occur in STZ diabetic animals [46,47]. These findings may be consistent with earlier observations [46,47] that STZ induction causes increases in blood triglycerides and total cholesterol levels. These substances are lower in SA-treated diabetic mice than those in vehicle-treated STZ mice (Figure 8).

PPARα agonists are available to lower the blood triglyceride levels [30]. Fenofibrate is one of the known PPARα agonists and is used to lower the triglyceride levels by mechanisms related to fatty acid oxidation [30]. PPARα agonists are known to downregulate numerous genes involved in lipid synthesis. Therefore, PPARα or target gene levels, including fatty acid synthase (FAS), were examined. Our findings showed that STZ induction caused a decrease in PPARα but an increase in the expression levels of FAS. Fatty acid synthase serves as a key enzyme in fatty acid synthesis [48]. Our findings showed that administration of SA, Glib, or Feno reduced blood TG levels. Perhaps the evidence that induction of hepatic PPARα receptor is responsible for increasing hepatic fatty acid oxidation is the observation that SA, Glib, or Feno lower blood TG levels by increasing hepatic β-oxidation activity and decreasing hepatic triglyceride synthesis. Moreover, PPARα-deficient mice displayed dysregulated SREBP-mediated lipogenic genes [30]. The role of SREBP1c as a lipogenic transcription factor may be to stimulate lipogenic enzyme expression and contribute to fatty acid synthesis and TG accumulation [49]. In this study, treatment with SA or Feno decreased the mRNA levels of lipogenic SREBP1c to decrease hepatic triglyceride output, apparently resulting in an antihypertriglyceridemic effect of SA or Feno. All of these are beneficial for ameliorating hepatic triacylglycerol accumulation within lipid droplets (for histological examination) and are absent in SA- or Feno-treated mice, which in turn leads to a decrease in blood TG levels.

A study reported that SREBP2 could play an important role in the regulation of cholesterol synthesis [49]. Our findings that STZ-induced mice treated with SA had decreased blood TC levels imply that cholesterol synthesis is inhibited in STZ-induced mice treated with SA. This is due to decreased SREBP2 mRNA in the liver, and TC synthesis is restored toward normal levels. The net effect would be to restore blood TC toward vehicle-treated STZ levels.

Taken together, the STZ-induced mice treated with SA could have normalized hyperlipidemia, including blood TG and TC levels, by a reduction in the expression levels of FAS and mRNA levels of SREBP2 but an increase in expression levels of PPARα in the liver.

PPARγ plays a key role in the stimulation of adipogenesis and lipogenesis [50]. Following treatment with SA, Glib, or Feno, a decrease in the expression levels of PPARγ may lead to the inhibition of adipogenesis and a reduction in lipid accumulation in liver tissues. Previous evidence has shown that the administration of globular domains of adiponectin may enhance glucose uptake and fatty acid oxidation, which is negatively associated with plasma lipid makers [51]. Leptin has been reported to play a key role in the regulation of adipocyte differentiation and adipose tissue metabolism [52]. In the present study, administration of SA2, SA3, Glib, or Feno to STZ-induced diabetic mice not only increased blood adiponectin and leptin concentrations (only SA3 with statistical significance) but also decreased FFA levels. The circulating lipids fluctuating throughout the body is of concern. Thus, the beneficial effect of SA on circulating lipids would be partly due to inhibition of hepatic lipid synthesis and enhancement of fat oxidation as well as regulation of adiponectin and/or leptin secretion, and SA acts in a dose-dependent manner.

The liver tissue plays a major role in lipid and lipoprotein metabolism, and the accumulated lipids in the adipose tissues are mostly from circulating TG [53], suggesting that SA may decrease adipogenesis and increase lipid catabolism in the liver including increased expression levels of PPARα but decreased FAS and reduced mRNA levels of SREBP1c, and, as a consequence, a reduction in hepatic lipid droplets (Figure 4A), which in turn lead to a decrease in plasma TG concentrations.

Following treatment with sulphurenic acid for 3 weeks in STZ-induced diabetic mice, we did not find any existing adverse effects. A piece of previous study was conducted to assess the 90-day oral toxicity of *A. cimamomea* from submerged culture in male and female Sprague-Dawley rats [54]. At the end of the experiment, no significant differences were found in urinalysis, hematology, and serum biochemistry parameters between the treatment and control groups [54]. Necropsy and histological examination indicated no treatment-related changes. According to the above results, the no-observed-adverse-effect level (NOAEL) of *A. cinnamomea* is identified to be greater than 3000 mg/kg BW/day in Sprague-Dawley rats [54]. In this study, the LD50 or TD50 of sulphurenic acid could not be estimated, and it needs to be clarified in the future.

## 4. Materials and Methods

### 4.1. Chemicals

Antibodies to GLUT4 (no. sc-7938) were purchased from Santa Cruz Biotech (Santa Cruz, Paso Robles, CA, USA); phospho-AMPK (Thr^172^), PPARα (no. ab8934), and PPARγ (no. ab45036) were obtained from Abcam Inc. (Cambridge, MA, USA); FAS (no. 3180), phospho-Akt (Ser^473^) (no. 4060), total AMPK (Thr^172^), phospho-FoxO1 (Ser^256^) (no. 11115), total-FoxO1 (Ser^256^) (no. 2880), and β-actin (no. 4970) were purchased from Cell Signaling Technology (Danvers, MA, USA). Secondary anti-rabbit antibodies were obtained from Jackson ImmunoRes. Lab., Inc. (West Grove, PA, West Baltimore Pike, USA). Anti-insulin (1 :100, no. sc-9168) or anti-glucagon (1 : 200, no. sc-13091) primary antibodies were obtained from Santa Cruz Biotechnology (Santa Cruz Inc., Paso Robles, CA, USA).

### 4.2. Isolation and Determination of the Active Compound

The freeze-dried powder of *Antrodia camphorata* submerged in whole broth was provided by the Biotechnology Center of Grape King Inc. (Chung-Li City, Taiwan). The powder of *A. camphorata* (1.6 kg) was extracted three times with methanol (16 L) at room temperature (4 days each). The methanol extract was evaporated in vacuo to obtain a brown residue, which was suspended in H_2_O (1 L) and then partitioned (three times) with 1 L of ethyl acetate. The EtOAc fraction (95 g) was chromatographed on silica gel and eluted with mixtures of hexane and EtOAc to increase polarity and then further purified with high-performance liquid chromatography (HPLC). Sulphurenic acid (24-methylenelanosta-8-ene-3β, 15α-diol-21-oic acid, 10; TR3; SA) (1.1 g) was eluted with 50% EtOAc in hexane and recrystallized with EtOH.

Sulphurenic acid: white powder; mp 246–248 °C (MeOH); (α)_D_ + 36.4 °C (c 0.22, MeOH); 1H NMR (300 MHz, pyridine-d_5_): δ 3.41 (1H, dd, *J* = 7.0, 8.3 Hz, H-3), 4.61 (1H, dd, *J* = 6.2, 8.9 Hz, H-15), 1.05 (3H, s, H-18), 1.05 (3H, s, H-19), 1.00 (3H, d, *J* = 6.9 Hz, H-26), 0.99 (3H, d, *J* = 6.8 Hz, H-27), 4.84 (1H, br s, H-28a), 4.87 (1H, br s, H-28b), 1.34 (3H, s, H-29), 1.20 (3H, s, H-30), 1.17 (3H, s, H-31); ^13^C NMR (75 MHz, pyridine-d5): δ 36.2 (*t*, C-1), 28.7 (*t*, C-2), 78.1 (*d*, C-3), 39.3 (*s*, C-4), 50.9 (*d*, C-5), 18.9 (*t*, C-6), 27.7 (*t*, C-7), 134.9 (*s*, C-8), 135.2 (*s*, C-9), 37.3 (*s*, C-10), 21.2 (*t*, C-11), 30.2 (*t*, C-12), 45.2 (*s*, C-13), 52.2 (*s*, C-14), 72.5 (*d*, C-15), 39.5 (*t*, C-16), 46.7 (*d*, C-17), 16.9 (*q*, C-18), 19.4 (*q*, C-19), 49.0 (*d*, C-20), 178.7 (*s*, C-21), 31.9 (*t*, C-22), 32.7 (*t*, C-23), 155.9 (*s*, C-24), 34.2 (*d*, C-25), 21.9 (*q*, C-26), 22.0 (*q*, C-27), 107.1 (*t*, C-28), 18.1 (*q*, C-29), 28.6 (*q*, C-30), 16.3 (*q*, C-31); APCI-MS (pos.): *m/z* 467 (M + 1) [55,56].

### 4.3. In Vitro Experiment

#### 4.3.1. Cell Culture

Myoblast C2C12 cells (ATCC, CRL-1772) were incubated in Dulbecco’s modified Eagle’s medium (DMEM, Gibco BRL) supplemented with 10% fetal bovine serum (FBS, HyClone), 100 U/mL penicillin and 100 μg/mL streptomycin under a humidified atmosphere with culture conditions set at 37 °C and 5% CO_2_. Cells for differentiation into myotubes were reseeded on 9-cm plates at a density of 1 × 10^5^ cells. After 48 h (over 80% confluence), the medium was switched to DMEM with 1% (*v*/*v*) FBS and was replaced after 2, 4, and 6 days of culture. Treatment of cells with 1, 5, 10, and 25 μg/mL SA, or 10 nM insulin was initiated on day 6 when myotube differentiation was complete, which was described in our previous report [57].

#### 4.3.2. Detection of Expression Levels of Phosphorylated Akt (Ser 473) in a Cell Line

The protein concentration of the supernatant was determined via the BCA assay (Pierce). Equal amounts of protein were then diluted 4× in SDS sample buffer (62.5 mM Tris-HCl, 20% glycerol, 2% SDS, 75 M DTT, and 0.05% bromophenol blue), subjected to SDS-PAGE and detected by Western blotting with antibodies specific for membrane total-Akt (t-Akt) and phospho-Akt (Ser^473^) (p-Akt) [57]. The density of the blot was analyzed using Alpha Easy FC software (Alpha Innotech Corp., Randburg, South Africa).

### 4.4. Animals and Treatments

The animal protocol was approved by the guidelines of Central Taiwan University of Science and Technology, Taiwan, in accordance with Institutional Animal Care and Use Committee (IACUC) (No. 106-CTUST-08), and approved by local animal ethics committee (Affidavit of Approval of Central Taiwan Institutional Animal Ethics Committee, permit #P106-I08) on 9, May, 2018. using 4-week-old male C57BL/6J mice obtained from the National Laboratory Animal Breeding Center. The study protocol for our animal treatment has been previously published [57]. Animals were maintained under a 12-h light: dark cycle and had free access to water and standard rodent chow. A group of CON mice were gavaged identical volumes of vehicle. Streptozotocin (Sigma Chemical, St. Louis, MO, USA) was dissolved in 0.05 M cold sodium citrate buffer (pH 4.5). After 7 days of acclimatization, diabetes was induced by daily 55 mg/kg intraperitoneal injections of streptozotocin solution over five days [57], followed by a one-week waiting period. STZ-induced mice were found to develop hyperglycemia, which was defined as fasting blood glucose above 250 mg/dL [58], and finally, only 42 mice were considered to have diabetes for this study. The STZ-induced diabetic micewere then randomly divided into six groups (*n* = 7 per group). Three groups were orally gavaged with SA at either 10, 20, or 40 mg/kg (groups SA1, SA2, and SA3, respectively). Three comparator groups were treated with similar volumes of either distilled water (STZ control group), glibenclamide (Glib) (10 mg/kg), or fenofibrate (Feno) (250 mg/kg). The vehicle, SA, Glib, and Feno were given by oral gavage once daily for 3 weeks.

For the assay of blood biochemical parameters, all mice were fasted for 10 h overnight, and blood samples (approximately 150–200 μL) were collected from the retro-orbital sinus under anesthesia. At the end of the experiment, mice were euthanized with carbon dioxide, and the liver, skeletal muscle, and white adipose tissues (WATs) (including epididymal, mesenteric and retroperitoneal WAT) were dissected and weighed and immediately frozen at −80 °C for target gene assays. Aside from the assay of blood glucose levels, heparin (30 units/mL) (Sigma) was first added to the blood collecting tubes, followed by drying of the tubes. The heparin-processed tubes were immediately used to collect blood samples. Plasma samples were collected by centrifugation of whole blood at 1600 *g* for 15 min at 4 °C, and this separation procedure was performed within 30 min. The supernatants were collected for biochemical analyses (including total cholesterol (TC) and triglyceride (TG), 20–30 μL). Aliquots of plasma samples (> 25 μL) were collected for insulin, adiponectin, and leptin assays. Body weight was measured daily (10:00 AM.) at the same time throughout the period. The amounts of pellet food were weighed, and then the amounts of remaining food were weighed after 24 h. The difference was regarded as the daily food intake [57]. Additionally, mouse monitoring was performed, including body weight changes, skin disease, food consumption, and the appearance of all mice.

### 4.5. Analysis of Blood Glucose, Biochemical, Adipocytokine, and Glycated Hemoglobin (HbA1_C_) Levels

Fasting blood glucose levels were determined by collecting blood samples (> 20 μL) drawn from retro-orbital sinuses. Approximately 20 μL of fresh blood sample was quickly placed on tinfoil, and the blood glucose level was read using a machine (Model 1500; Sidekick Glucose Analyzer; YSI Incorporated, Yellow Springs, OH, USA) by the glucose oxidase method. Blood TG and TC levels were determined using commercial assay kits in accordance with the manufacturer’s directions (Triglycerides-E test and Cholesterol-E test, Wako Pure Chemical, Osaka, Japan). Aliquots of plasma samples (> 25 μL) were collected for insulin, adiponectin, and leptin assays using enzyme-linked immunosorbent assay (ELISA) kits (mouse insulin ELISA kit, Mercodia, Uppsala, Sweden; adiponectin ELISA kit, Crystal chem; mouse leptin ELISA kit, Morinaga, Yokohama, Japan) as described in previous studies [57,58]. The percentage of HbA1c was determined using a Hemoglobin A_1C_ kit (BioSystems S.A., Barcelona, Spain).

### 4.6. Histological Examination

Small pieces of liver tissues and pancreas were fixed with formalin (200 g/kg) neutral buffered solution and embedded in paraffin. Sections (8 µm) were cut and stained with hematoxylin and eosin. For microscopic examination, a microscope (Leica, DM2500) was used, and the images were taken using a Leica Digital camera (DFC-425-C) at 10 (ocular) × 10 (object lens) magnification. Each presented image is typical and representative of seven mice.

An immunohistochemical (IHC) staining for insulin (brown) and glucagon (green) in the pancreatic islets of mice was assessed according to a previous study [59], and briefly, anti-insulin (1 : 100, Santa Cruz Biotechnology, Inc., Santa Cruz, CA, USA, no. sc-9168) or anti-glucagon (1 : 200, Santa Cruz Biotechnology, no. sc-13091) primary antibodies were used. Staining was developed using Histostain-Plus Broad Spectrum (AEC) Kit (Invitrogen, Frederick, MD, USA, no. 859943). The IHC procedure was conducted according to manufacturer instructions, and image were taken at 400 magnification.

### 4.7. Relative Quantification of mRNA and Western Blotting

Relative quantification of mRNA (primers are shown in Table 1) and immunoblots for assessment of the expression levels of membrane GLUT4 in the skeletal muscle and hepatic phospho-AMPK (Thr^172^) and phospho-Akt were performed as described in previously reported studies [6,57]. The liver tissues were assessed for the expression levels of phospho-Akt, phospho-FoxO1, PPARα, FAS, and PPARγ. Skeletal muscle from mice was subjected to the measurement of GLUT4 expression levels, and the total membrane fraction was collected and assessed as previously described [60,61]. Skeletal muscle was powdered under liquid nitrogen and homogenized in buffer (pH 7.4) containing 250 mmol/L sucrose, 50 mmol/L Tris, and 0.2 mmol/L edetic acid for 20 s. The homogenate was centrifuged at 9000× *g* for 10 min (4 °C), and the supernatant was removed. The pellets were cleaned with buffer and centrifuged three times. All three supernatants were mixed and centrifuged at 190,000× *g* for 60 min (4 °C). The resulting pellet was resuspended in a small amount of buffer (approximately 0.5 mL) as a total membrane fraction [60,61]. The membrane expression levels of GLUT4, phospho-AMPK, and total AMPK were analyzed by Western blotting as described in previously reported studies [6,57].

### 4.8. Statistical Analysis

All of the results are presented as the mean and standard error. Whenever needed, data were subjected to analysis of variance, followed by Dunnett’s multiple range tests, using SPSS software (25.0.0.0, SPSS Inc., Chicago, IL, USA). *p* < 0.05 was considered to be statistically significant.

## 5. Conclusions

The present study was designed to assess the potential effects of SA, a pure compound, in in vitro and in vivo experiments. In an in vitro study, SA enhanced Akt phosphorylation in the absence of insulin in C2C12 myotubes, implying that SA had an action similar to that of insulin. Animal studies using SA in STZ-induced diabetic mice indicated that the SA compound displayed an antihyperglycemic effect that had a great glucose-lowering effect by enhancing the size of the islets of Langerhans cells. Treatment with SA enhanced the expression levels of phosphorylated Akt and activation of AMPK and membrane GLUT4 to increase glucose uptake in the skeletal muscles. On the other hand, SA enhanced the hepatic expression levels of p-Akt and p-FoxO1 but reduced the mRNA levels of PEPCK and G6Pase to inhibit hepatic glucose production. The overall effect accounts for the antidiabetic activity of SA in type 1 diabetic mice. SA treatment can increase the phosphorylation levels of AMPK and enhance the expression levels of PPARα in the liver but decrease lipogenic FAS and reduce the mRNA levels of SREBP 2, thus leading to a decrease in blood TG and TC. SA may alleviate type 1 diabetes symptoms, or may be used in combination with insulin, but it will not be a sole treatment.

## Figures and Tables

**Figure 1 ijms-20-04897-f001:**
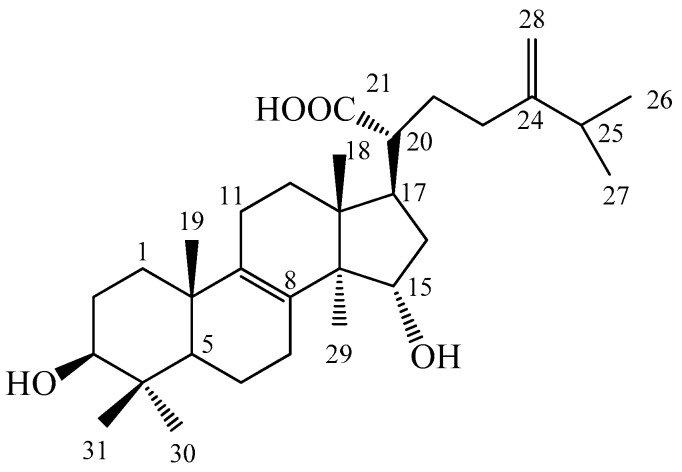
Chemical structure of sulphurenic acid (SA).

**Figure 2 ijms-20-04897-f002:**
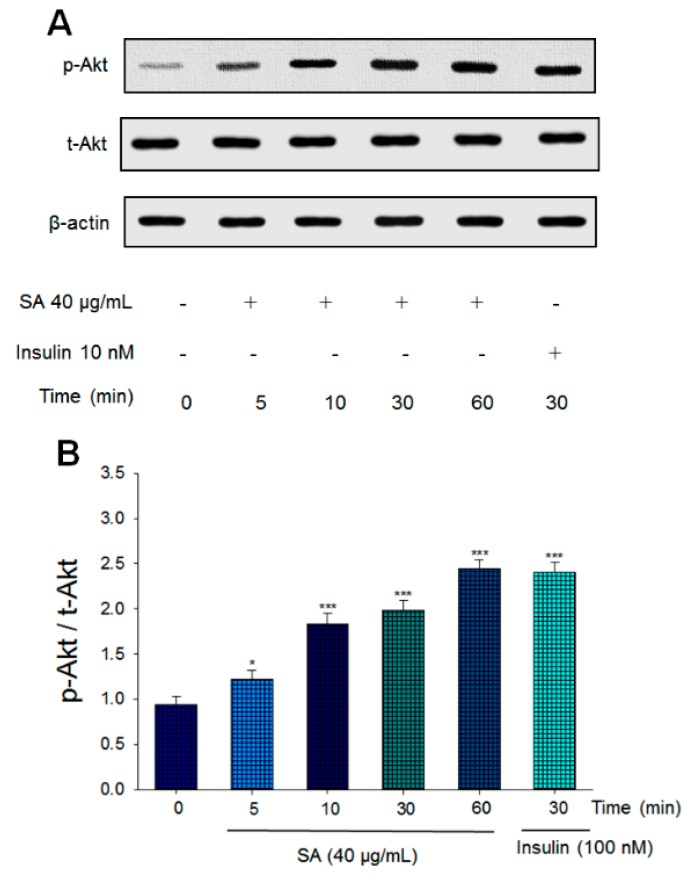
Sulphurenic acid (SA) activates Akt signaling pathways. The cell lysates were analyzed via Western blotting for phospho-Akt (p-Akt) and total-Akt (t-Akt). (**A**) Representative image; Akt phosphorylation was determined from C2C12 cells, and treated with 40 μg/mL of SA for the indicated period of time (5–60 min); (**B**) The ratios of p-Akt to t-Akt forms were analyzed and presented as phosphorylation of Akt. * *p* < 0.05, *** *p* < 0.001 compared with the 0 min group. All values are means ± SE (*n* = 3).

**Figure 3 ijms-20-04897-f003:**
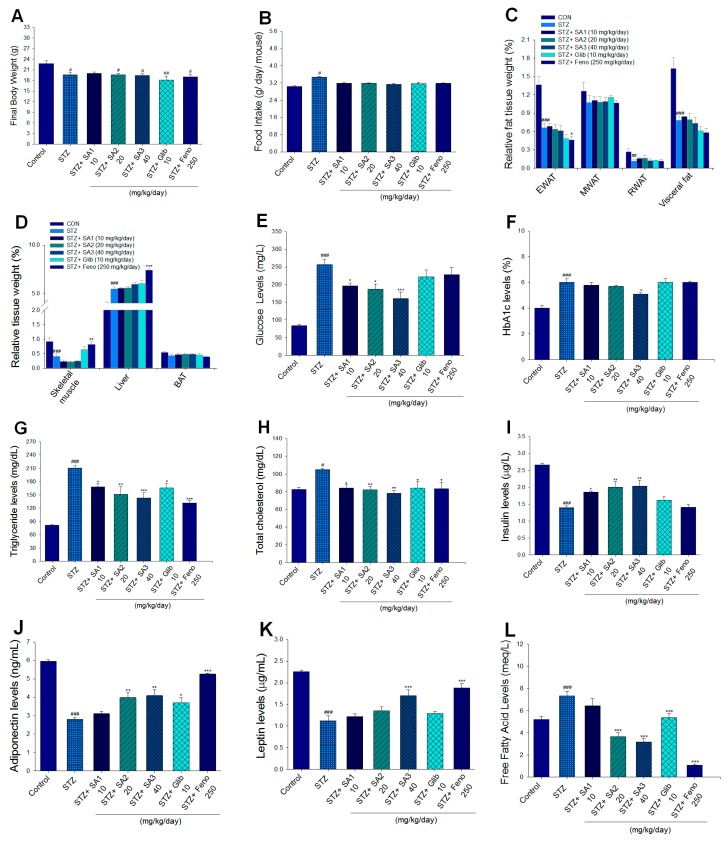
Effects of sulphurenic acid (SA) on (**A**) final body weight, (**B**) food intake over 3-week treatment, (**C**) relative fat tissue weight, (**D**) relative tissue weight, (**E**) blood glucose levels, (**F**) blood glycated hemoglobin (HbA1C) levels, (**G**) triglyceride levels, (**H**) total cholesterol levels, (**I**) insulin levels, (**J**) adiponectin levels, (**K**) leptin levels, and (**L**) free fatty acid levels in streptozotocin-induced diabetic mice. ^#^
*p* < 0.05, ^##^
*p* < 0.01, and ^###^
*p* < 0.001 compared with the control (CON) group; * *p* < 0.05, ** *p* < 0.01, and *** *p* < 0.001 compared with the streptozotocin (STZ) plus vehicle (distilled water) (STZ) group. All values are means ± SE (*n* = 7 per group). Sulphurenic acid (SA): SA1: 10, SA2: 20, SA3: 40 mg/kg body weight; Glib: glibenclamide (10 mg/kg body weight); Feno: fenofibrate (250 mg/kg body weight). EWAT, epididymal white adipose tissue; MWAT, mesenteric white adipose tissue; RWAT, retroperitoneal white adipose tissue; visceral fat is defined as EWAT+RWAT. RWAT, retroperioneal white adipose tissue; MWAT, mesenteric white adipose tissue.

**Figure 4 ijms-20-04897-f004:**
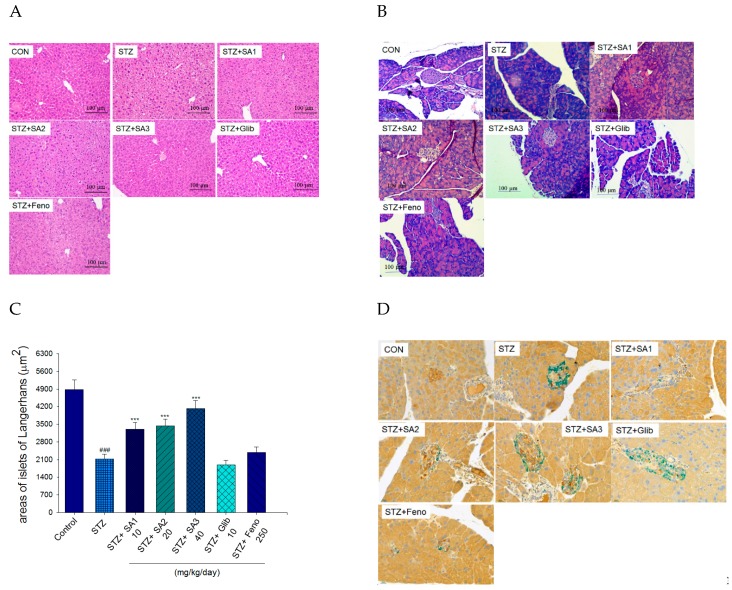
Histology examinations. (**A**) Liver tissues and (**B**) pancreatic islets of Langerhans, (**C**) the average area of islets of Langerhans, ^###^
*p* < 0.001 compared with the control (CON) group; *** *p* < 0.001 compared with the streptozotocin plus vehicle (distilled water) (STZ) group. (**D**) scale bar: 400×. an insulin (brown) and glucagon (green) immunohistochemical staining of pancreatic islets of mice in the control (CON), streptozotocin plus vehicle (distilled water) (STZ), STZ + SA1, STZ + SA2, STZ + SA3, STZ + glibenclamide (Glib), or STZ + fenofibrate (Feno) groups (*n* = 7 per group) by hematoxylin and eosin-staining. Sulphurenic acid (SA): SA1: 10, SA2: 20, SA3: 40 mg/kg body weight; Glib: glibenclamide (10 mg/kg body weight); Feno: fenofibrate (250 mg/kg body weight).

**Figure 5 ijms-20-04897-f005:**
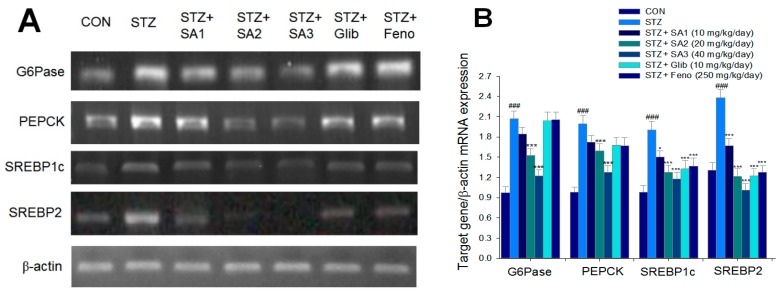
Semiquantative reverse transcription-polymerase chain reaction (RT-PCR) analysis on targeted gene mRNA levels in liver tissue of the mice by oral gavage sulphurenic acid (SA) (SA1, SA2, and SA3, 10, 20 and 40 mg/kg body weight), glibenclamide (Glib; 10 mg/kg body weight), or fenofibrate (Feno; 250 mg/kg body weight). (**A**) Representative image; (**B**) quantification of the ratio of target gene to β-actin mRNA expression. Total RNA (1 μg) isolated from tissue was reverse transcripted by MMLV-RT; 10 μL of RT products were used as templates for PCR. The expression levels of G6Pase, PEPCK, SREBP1c, and SREBP2 mRNA were measured and quantified by image analysis. Values were normalized to β-actin mRNA expression. All values are means ± SE (*n* = 7 per group). ^###^
*p* < 0.001 compared with the control (CON) group; *** *p* < 0.001, * *p* < 0.05 compared with the streptozotocin plus vehicle (distilled water) (STZ) group.

**Figure 6 ijms-20-04897-f006:**
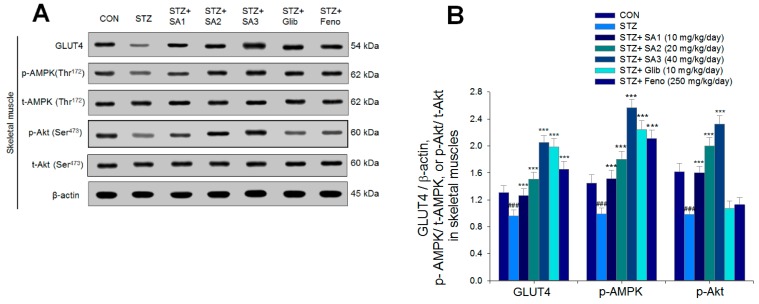
The expression levels of (**A**) membrane GLUT4, p-AMPK (Thr^172^)/t-AMPK, and p-Akt (Ser^473^)/t-Akt (Ser^473^) in the skeletal muscles of STZ-induced diabetic mice by oral gavage sulphurenic acid (SA). (**A**) Representative image; (**B**) quantification of the p-AMPK to t-AMPK and p-Akt (Ser^473^)/t-Akt (Ser^473^). Protein was separated by 12% SDS-PAGE detected by Western blot. ^###^
*p* < 0.001 compared with the control (CON) group; *** *p* < 0.001 compared with the streptozotocin plus vehicle (distilled water) (STZ) group. All values are means ± SE (*n* = 7 per group). SA1, SA2, and SA3, sulphurenic acid (SA) (SA1, SA2, and SA3, 10, 20, and 40 mg/kg body weight, respectively); glibenclamide (Glib, 10 mg/kg body weight); fenofibrate (Feno; 250 mg/kg body weight).

**Figure 7 ijms-20-04897-f007:**
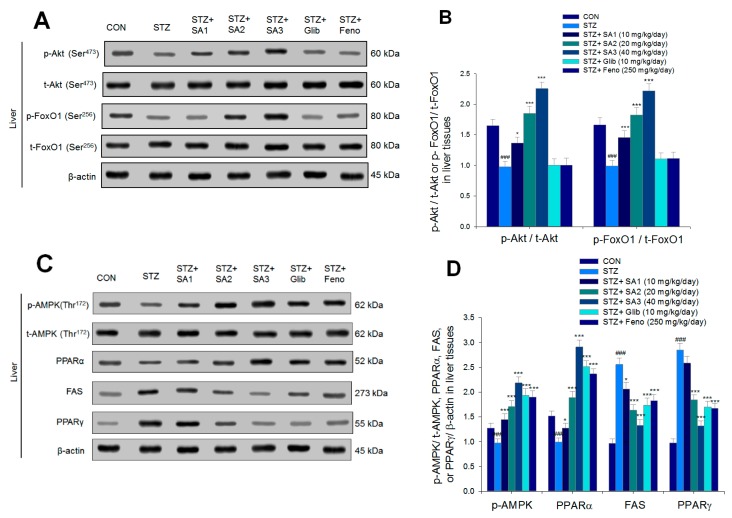
The expression levels of (**A**) p-Akt (Ser^473^)/t-Akt (Ser^473^) and p-FoxO1 (Ser^256^)/t-FoxO1 (Ser^256^), (**B**) p-AMPK (Thr^172^)/t-AMPK, PPARα, FAS, and PPARγ in the liver tissues of STZ-induced diabetic mice by oral gavage sulphurenic acid (SA). (**A**,**B**) Representative image; (**C**,**D**) quantification of the p-AMPK to t-AMPK, PPARα, FAS, and PPARγ. Protein was separated by 12% SDS-PAGE detected by Western blot. All values are means ± SE (*n* = 7 per group). ^###^
*p* < 0.001 compared with the control (CON) group; * *p* < 0.05, *** *p* < 0.001 compared with the streptozotocin plus vehicle (distilled water) (STZ) group. SA1, SA2, and SA3, sulphurenic acid (SA) (SA1, SA2, and SA3, 10, 20, and 40 mg/kg body weight, respectively); glibenclamide (Glib, 10 mg/kg body weight); fenofibrate (Feno; 250 mg/kg body weight).

**Figure 8 ijms-20-04897-f008:**
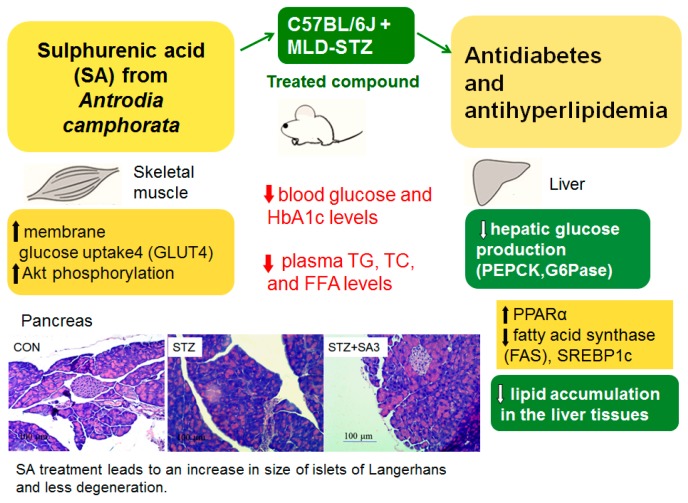
The graphical abstract of sulphurenic acid (SA) in multiple low dose (MLD) streptozotocin (STZ)-induced diabetic mice.

**Table 1 ijms-20-04897-t001:** Primers used in the present study.

Gene	Accession Number	Forward Primer and Reverse Primer	PCR Product (bp)	Annealing Temperature (°C)
Liver				
*PEPCK*	NM_011044.2	F: CTACAACTTCGGCAAATACC R: TCCAGATACCTGTCGATCTC	330	52
*G6Pase*	NM_008061.3	F: GAACAACTAAAGCCTCTGAAAC R: TTGCTCGATACATAAAACACTC	350	50
*SREBP1c*	NM_011480	F: GGCTGTTGTCTACCATAAGC R: AGGAAGAAACGTGTCAAGAA	219	50
*SREBP2*	AF289715.2	F: ATATCATTGAAAAGCGCTAC R: ATTTTCAAGTCCACATCACT	256	47
*β-actin*	NM_007392	F: TCTCCACCTTCCAGCAGATGT R: GCTCAGTAACAGTCCGCCTAGA	99	55

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
