# Peer review of "Antidiabetic and Antihyperlipidemic Effects of Sulphurenic Acid, a Triterpenoid Compound from Antrodia camphorata, in Streptozotocin-Induced Diabetic Mice"

_ijms, 2019, doi:10.3390/ijms20194897_

Round 1

Reviewer 1 Report

The manuscript describes the antidiabetic effects of a triterpenoid compound in glucose and lipid parameters using a mouse model of type 1 diabetes. Below some comments that may improve the manuscript further.

In the histological analysis of pancreas, an immunostaining for insulin should be performed. This will confirm that STZ treatment worked, and will strengthen the hypothesis that SA improves insulin secretion. Also, the claim that islets are larger and in higher number in an experimental group should be backed up by size measurements reported in a graph.

How can the discrepancy between the molecular size of AMPKa total and phosphorylated in figure 6 can be explained?

HbA1c usually does not change until 5 weeks of treatment in mice. Hence, the remarkable increase in the STZ group of this circulating marker is odd and concerning. It should be further explained/discussed in the discussion section.

Assessment of oxidative stress, either by measurement of glutathione peroxidase, catalase or superoxide dismutase activity is also highly recommended in the pancreas to strengthen further the antidiabetic claim being made in the manuscript.

The claim in line 371 that SA may be used as treatment for type 1 diabetes is very far-fetching. SA may alleviate type 1 diabetes symptoms, or may be used in combination with insulin. Claorify that it will not be a sole treatment is necessary.

Regarding the English language, there are awkward sentences, as well as a few typos and run-on sentences. (Too many to point out discreetly here.) A language review by either a native speaker or a professional service would improve significantly the quality of this article.

Minor comments:

L.52 – “Precious” is not an appropriate adjective in this context.

L.56 – “Physiological functions” is a tautology. Please correct.

L.85-88 – Unclear sentence.

L.97-98 – Clarify whether fenofibrate is an agonist or antagonist of PPARa.

Section 2.1 – Please clarify that experiment was performed in a cell line.

L.119 – “Body weight AVERAGE of all mice”?

L.120 – What is CON group? Define every acronym the first time it appears.

L.446 – It is concerning that an n=7 for the STZ group was further divided into six groups. Is it n=7 per group? If that, n value for the STZ group needs to be corrected. The same correction should be made in the figure 3 legend.

Figure 4 – Legend lacks the n value for each group.

Section 4.7 – “Relative quantiFICation”

Section 4.8 – Whenever possible, or whenever needed?

L.519 – “STZ-INDUCED diabetes”

Reviewer 2 Report

This paper presents data on the antidiabetic and antihyperlipidemic effects of sulphurenic acid – a triterpenoid obtained from Antrodia camphorate in mice with streptozotocin-induced diabetes. The study is interesting and brings novelty. The experiments seems to be well-planned and conducted. However, several issues need to addressed by the Authors:

The manuscript should be carefully checked for style and grammar, preferably by a native speaker or someone with proficiency in English. The abbreviations should be used more consistently. Are there any adverse effects following the long-term treatment with sulphurenic acid? The toxicity issue of sulphurenic acid should be mentioned and discussed accordingly. Was the LD50 or TD50 dose estimated? The Conclusions section is a bit too long.

Round 2

Reviewer 2 Report

I have no further comments.